# Alloy Partitioning Effect on Strength and Toughness of κ-Carbide Strengthened Steels

**DOI:** 10.3390/ma15051670

**Published:** 2022-02-23

**Authors:** Daniel M. Field, Krista R. Limmer, Billy C. Hornbuckle, Dean T. Pierce, Ken E. Moore, Katherine M. Sebeck

**Affiliations:** 1US Army Combat Capabilities Development Command Army Research Laboratory, Aberdeen Proving Ground, Aberdeen, MD 21005, USA; krista.r.limmer.civ@army.mil (K.R.L.); billy.c.hornbuckle.civ@army.mil (B.C.H.); 2Materials Science and Technology Division, Oak Ridge National Laboratory, Oak Ridge, TN 37830, USA; piercedt@ornl.gov (D.T.P.); mooreke@ornl.gov (K.E.M.); 3US Army Combat Capabilities Development Command Ground Vehicle Systems Center, Warren, MI 48092, USA; katherine.m.sebeck.civ@army.mil

**Keywords:** κ-carbide, age-hardening steel, alloy partitioning, low density steel

## Abstract

Alloy partitioning during heat treatment in a lightweight precipitation hardened steel was investigated using transmission electron microscopy and atom probe tomography. The mechanical properties are discussed as a function of the effect of solution treatment temperature and aging time, giving rise to variations in chemical modulation. A wrought lightweight steel alloy with a nominal composition of Fe-30Mn-9Al-1Si-1C-0.5Mo (wt. %) was solution-treated between 1173–1273 K and aged at 773 K. Lower solution treatment temperatures retained a finer grain size and accelerated age hardening response that also produced an improved work hardening behavior with a tensile strength of −1460 MPa at 0.4 true strain. Atom probe tomography indicated these conditions also had reduced modulation in the Si and Al content due to the reduced aging time preventing silicon from diffusing out of the κ-carbide into the austenite. This work provides the framework for heat-treating lightweight, age hardenable steels with high strength and improved energy absorption.

## 1. Introduction

Weight and fuel efficiency have become a pivotal concern for vehicle designers. Lightweight steels have become one avenue for reducing weight with equivalent strength and energy absorbing capability as traditional quench and tempered steels [1,2]. Low density austenitic steels, termed FeMnAl steels, contain high concentrations of Al, Mn, and C [1,2,3,4,5,6,7,8,9,10,11]. Age hardening to produce κ-carbide is typically performed within the temperature range of 673–973 K [4,6]. Prior investigations have primarily focused on either investigating the isothermal aging response of a single alloy composition or exploring the effect of composition at a fixed aging condition [1,2,3,4,5,6,7,8,9]; little is known about the mechanical response and underlying mechanisms for a single alloy processed through variable thermal treatments to the same nominal strength. Lu et al. [3] investigated the microstructural response of a Fe-26Mn-9Al-0.75C (in wt. pct.) alloy age hardened after cold rolling; aging was performed for a fixed time of 6hr at temperatures ranging from 773 to 1173 K. After 6hr of aging, they noted the formation of κ-carbide from the eutectoid decomposition of γ-austenite (γ → κ + α) within the temperature range of 873 to 1073 K. Lu et al. also showed that above 1073 K no κ-carbide was observed and theorized that this temperature is the solvus for κ-carbide.

κ-carbide formation has been evaluated as a function of composition with a fixed time at temperature by several authors. Bartlett et al. investigated the effect of silicon [8] and phosphorous [9] on the age hardening response of cast FeMnAl alloys. They reported Si and P additions individually accelerate the aging hardening kinetics of these low-density steels. In Bartlett et al.’s investigation on silicon, the increase in shear strength from age hardening was estimated according to the chemical modulation measured from atom probe tomography (APT). They showed by increasing the silicon content of the alloy and there was an increase in the chemical modulation of carbon between κ-carbide and austenite. This produced a greater increase in strength after aging for a fixed time and temperature of 60hr and 803 K; however the effect of aging time or temperature were not independently investigated. Yoo et al. [10] examined the effect of grain size on the mechanical properties of an as-solution treated FeMnAl steel at varying temperatures from 973 to 1273 K. They showed that the optimal work hardening rate to achieve the best strength-ductility combination (product of UTS and total elongation) occurred for the largest grain size. As the grain size increased from 5 to 38 μm, the work hardening rate reached a consistent maximum (−2000 MPa), irrespective of grain size. However, the coarsest grained sample demonstrated the maximum in work hardening at an increased strain level, leading to greater total elongation, thus identifying it as the better performing steel. Field and Limmer [12] evaluated a wrought alloy with nominal composition of Fe-30Mn-9.0Al-1.0C-0.5Mo (in wt. %) by varying the solution treatment temperature from 1223 to 1323 K and the subsequent aging time from 5 to 120hr at a fixed temperature of 773 K. They found after solution treating the alloy at 1223 K the toughness, as measured by Charpy v-notch impact energy, was improved. Furthermore, this increased toughness was retained at longer aging times compared to the alloy aged at 1323 K. The underlying reason for the improvement was attributed to the retention of a finer grain structure and the reduction in normal grain growth from the high temperature solution treatment. However, this study did not investigate the role of κ-carbide on toughness or strength.

Choi et al. [11] showed the effect of κ-carbide precipitation on the work hardening response of a Fe-28Mn-9Al-0.8C (in wt. pct.) steel; they discovered for solution-treated allows or those in under-aged condition the instantaneous work hardening exhibited a continuous increase up to 0.1 true strain. In the over-aged condition, it was found that the work hardening exhibited a continuous decrease. Choi theorized it was due to two root causes. First, for the over-aged condition the microstructure consisted of hard grain boundary particles and a relatively soft austenitic matrix leading to a behavior similar to that seen in a dual-phase (ferrite/martensite) steel as described by Kim et al. [13]. Second, the similar work hardening behavior between the under-aged and solution-treated condition was believed to be caused by the weak dislocation barrier associated with nanoscale κ-carbide. This was also confirmed by the work of Kim et al., which [14] showed that shearing was the operating mechanism when κ-carbide had a radius < 13 nm, and Orowan strengthening due to dislocation bypass for larger radius carbides.

All previous works have looked at the effect of either a fixed time of aging with varying age hardening temperature or a fixed aging condition (time and temperature) for a variation of compositions. This work is novel in that it will investigate the interacting effects of solution treatment of a single alloy on the mechanical behavior related to the compositional response of the alloy at the nano-carbide level.

## 2. Materials and Methods

A lightweight steel with a composition of Fe-30.8Mn-9.2Al-0.7Si-1.0C-0.5Mo (in wt. pct.) was industrially cast into a 5450 kg ingot. The stacking fault energy (SFE) of the alloy was calculated to be 89 mJ/m^2^ using a regular solution model according to the work of Pisarik and Van Aken [15,16]. After cooling the ingot was forged at 50%, and the forged slab was reheated and rolled to a plate thickness of 12.7 mm to obtain a total thickness reduction of 95%. The composition of the steel was measured using inductively coupled plasma optical emission spectroscopy (ICP-OES) after dissolution in nitric acid, and the carbon content was determined using gas combustion analysis.

Samples for solution treatment were placed in stainless steel bags, to minimize decarburization and oxidation, and heated to the solution treating temperature for 2hr followed by quenching in agitated room temperature water. This solution treatment and quenching process will hereafter be referred to as “STQ”. The temperature of the sample was tracked using an external thermocouple and heat treatment time was based upon surface temperature reaching the targeted treatment value. Heating rates ranged from 30 to 35 K/min. Age hardening was performed at 773 K for times up to 360hr followed by air cooling to room temperature.

Multiple hardness values were measured by Rockwell C and B techniques and converted to Brinell hardness (HBW) for reporting according to ASTM E18-17 [17]. The 1223, 1273 and 1323 K solution treatment followed by a 66hr age hardening treatment were selected to investigated the effect of a fixed aging time on microstructure and material behavior. The 1173 K STQ treatment was followed by 10hr age hardening to compare the effect of a variable age hardening time to a fixed strength. Standard ASTM E8 [18] tensile bars with a gauge length of 50 mm and gauge width of 12.5 mm were water-jet cut parallel to the rolling direction and the gauge edge was sanded to remove chatter marks from water-jet cutting. Triplicate testing was performed to obtain standard deviation values in yield, ultimate strength, and true strain. Gauge surfaces were ground to remove any surface oxide and decarburization from hot rolling and heat treatment. Strain was measured using digital image correlation (DIC) and tests were performed in displacement control to produce a strain rate of 4.0 × 10^−4^ s^−1^ using a 250 kN load cell on an Instron (Instron Co., Norwood, MA, USA) model 5989 frame.

Microscopy samples were mechanically polished to 0.02 μm with colloidal silica solution and etched with 10% nital solution of nitric acid and methanol. Scanning electron microscopy (SEM) images were taken with a Phenom XL (Thermo Fisher Scientific, Waltham, MA, USA) operating at 15 kV using the backscatter mode. Specimens for electron backscattered diffraction (EBSD) were observed prior to etching. Orientation image mapping via pattern analysis was performed on an FEI NanoSEM (Thermo Fisher Scientific, Waltham, MA, USA) operated at an accelerating voltage of 20.0 kV and an emission current of 2.7 nA.

Samples for transmission electron microscopy were prepared by twin-jet electropolishing (Struers Inc., Cleveland, OH, USA) (Tenupol-5) and also by focused ion beam (FIB) lift-out. The electropolishing was completed using a 70% methanol/30% nitric acid solution at −30 °C. FIB samples were taken from grain boundaries using a ThermoFisher Scientific Thermo Fisher Scientific, Waltham, MA, USA) Helios G4 UX dual-beam focused ion beam (FIB) / scanning electron microscope (SEM). The FIB was operated at 30 keV with maximum and minimum beam currents of 0.75 nA and 90 pA, respectively. To minimize Ga implantation into the tip, the final step was conducted at 5 keV with a beam current of 21 pA. Grain boundaries with character angles ranging from 15 to 55 degrees were selected for lift out (determined by electron backscatter diffraction), as this range of character angle encompassed the majority of grain boundaries in the microstructure, excluding twin boundaries. Scanning Transmission Electron Microscopy (STEM) was performed on the samples using a Hitachi (Hitachi High-Tech, Schaumburg, IL, USA) HF-3300 operating at 200 kV.

The atom probe tomography (APT) analysis was performed utilizing a Cameca (CAMECA, Gennevilliers, France) Local Electrode Atom Probe (LEAP) 5000XR system. Atom probe specimens were prepared using an in-situ lift-out method then annularly ion milled to the appropriate geometry with a Thermo Scientific Helios G4 dual beam microscope. The specimens were run in laser mode with a pulse rate of 250 kHz, pulse energy of 25 pJ, and a target evaporation rate of 0.5% while maintaining a base temperature of 50 K. A minimum of 25 million ions were collected for each tip analyzed, and two tips were analyzed for each condition. The collected datasets were reconstructed using Cameca’s Imago Visualization and Analysis Software (IVAS) version 3.8.4 software (Gennevilliers, France).

## 3. Results

The age hardening behavior of the alloy at 773 K with different preceding STQ treatments is shown in Figure 1. Earlier work with this alloy has shown that the STQ hardness is a function of grain size with rapid grain growth at increasing STQ temperatures [19]. The hardening rate and time-dependent hardness increased as the STQ temperature was decreased. The 1323 K STQ sample does not show a hardening response until aging is performed for more than 20hr at 773 K, whereas the 1173 K STQ sample exhibits an increase in hardness of 50 HBW after aging for 4 h.

One aged condition was selected for each of the four STQ treatments for further evaluation. Using a target hardness of 340 HBW, the samples were prepared by aging at 773 K for 10hr following an 1173 K STQ or 66hr following STQ at 1223, 1273, and 1323 K. 1173-10hr and 1273-66hr with the same nominal hardness, 1223-66hr hardest, and 1323-66hr—significantly softer. A representative true stress-true strain response of the alloy after aging at the selected times is shown in Figure 2a. The instantaneous work hardening rates were calculated according to Equation (1) and are shown in Figure 2b after logarithmic smoothing was applied to the stress-strain curve.
K = d*σ*/d*ϵ*(1)

The materials aged 66hr (1223, 1273, and 1323 K STQ condition) exhibit very similar work hardening trends, independent of the yield strength of the alloy, with variations only arising from the onset of necking and failure. However, the 1173-10hr condition exhibits a higher work hardening rate throughout plastic deformation. The yield strength of the 1173-10hr condition is similar to the 1273-66hr condition. However, the strain to failure, tensile strength, and instantaneous work hardening rate is higher in the 1173-10hr compared to the 1273-66hr condition. The tensile strength of the 1173-10hr condition is the same as the 1223-66hr condition, although the 1223-66hr condition had a significantly higher yield strength. As expected, the 1323-66hr condition had the lowest yield and tensile strengths and the greatest true strain. The tabulated mechanical properties are shown in Table 1, with values taken from previously reported results of the STQ conditions as published in ref. [19].

Scanning electron micrographs of the STQ and aged conditions are shown in Figure 3. Grain size was correlated to STQ temperature as has been reported previously by the authors [19]. There is no significant grain growth that occurs during the aging process to differentiate the grain size of the aged material compared to the solution treated samples. The eutectoid decomposition of γ-austenite to α-ferrite + κ-carbide was not observed in this alloy system. The orientation image maps (OIM) generated from EBSD of the tensile tested 1173-10hr and 1323-66hr samples are shown in Figure 4. As is expected due to the alloy’s high SFE, strain is accommodated by slip, not twinning or phase transformation. Slip bands are indicated with white arrows and the annealing twins, not mechanical twins, are identified by yellow arrows. It should be noted that the annealing twin boundaries are no longer parallel due to the deformation applied during tensile testing, and boundaries appear bent or bowed.

Transmission electron microscopy (TEM) was performed to identify κ-carbide as well as determine the orientation relationship between the κ-carbide and the austenite. A representative TEM micrograph of the 1223-66hr sample is shown in Figure 5a with the chemical modulations observed in the grain interior of the parent austenite. Selected area diffraction (SAD) pattern of the <001> zone axis is shown in Figure 5b. The austenite reflections are identified and the superlattice <001> κ-carbide reflections are shown to be parallel to the <002> austenite, producing the typical cube on cube orientation relationship.

The grain boundaries were further investigated to determine if κ-carbide would precipitate at grain boundaries as the grain boundary area increased (finer grain size). The 1173-10hr sample was examined using TEM, and a low angle grain boundary was observed as shown in Figure 6. The spinodal modulation was measured in Figure 6a to be about 2–3 nm. A higher resolution image of the grain boundary is shown in Figure 6b with the grain boundary being approximately 10 nm in width. It also noted that the grain boundaries appear to be free of modulation or κ-carbide.

Atom probe tomography was performed on the 1173 K-10hr, 1223 K-66hr, and 1273 K-66hr tensile bar grip sections. Atom maps and isoconcentration surfaces for C and Si are shown in Figure 7, complete atom maps for all major alloying elements are included within the appendix (Figure A1, Figure A2, Figure A3 and Figure A4). Carbon and aluminum enriched zones are used to identify κ-carbide. Carbon enriched zones are readily observed in the atom maps for both the 1273-66hr and 1223-66hr conditions (Figure 7f,k), whereas for the 1173-10hr condition the chemical modulation is not sufficiently well defined to be visible in the atom map (Figure 7a). Carbon isoconcentration surfaces generated at 5.6 at. % C (Figure 7b,g,l) were used to qualitatively visualize the size and morphology of the κ-carbide for each condition. The 1223-66hr condition has the largest κ-carbide and the longest spinodal decomposition wavelength, followed by the 1273-66hr condition, with the 1173-10hr condition having the smallest κ-carbide size and a mostly uniform distribution with no regular decomposition wavelength. Silicon is observed to be rejected from the κ-carbide, which is consistent with prior reports in the literature for aged alloys [8,9,20]. Silicon modulation is apparent in the Si atom maps for the 1223-66hr and 1273-66hr conditions (Figure 7h,m) whereas no apparent modulation is observed for the 1173-10hr condition (Figure 7c). Silicon isoconcentration surfaces generated at 2 at. % Si (Figure 7d,i,n) were used to qualitatively visualize the morphology of the austenite matrix. The 1223-66hr condition has the highest Si isoconcentration threshold (2.27 at. %) and the widest bands. The 1173-10hr and 1273-66hr conditions have comparable Si thresholds (1.9 at. %), although the 1273-66hr condition has more distinct modulation, with the 1173-10hr conditions having only a few small regions below the Si isoconcentration threshold. An overlay of the C and Si isoconcentration surfaces clearly demonstrates the separation between C and Si enriched regions for the 1223-66hr and 1273-66hr conditions (Figure 7j,o), whereas alignment is less apparent for the 1173-10hr condition (Figure 7e).

Proximity histograms (proxigrams) were used to quantify the composition of the κ-carbide and austenite. The interface is defined as 3.8% C and 4.02% C for 1173-10hr samples, 3.75% C and 3.8% C for the 1223-66hr samples, and 4.0% C for both 1273-66hr samples. The proxigrams shown in Figure 8 display the C, Si, Al, Mn, Fe, and Mo concentration profiles for the three different aged conditions. The noise evident in the 1173-10hr condition beyond ±1.5 nm from the interface is the result of the κ-carbide phase being on the order of 3 nm in size, such that relatively few atoms were collected that were more than 1.5 nm from the specified C-isosurface concentration. As was qualitatively observed in the atom maps and isoconcentration surfaces, the κ-carbide is significantly enriched in C for all three aging conditions. The partitioning behavior of Si and Al are dependent upon heat treatment conditions, with significant segregation of Si to austenite and Al to κ-carbide for longer aging times of 66hr, compared to the relatively unchanged Si and Al levels across the γ/κ interface after 10hr of aging. The lack of Al and Si modulation is proposed to be due to the shorter aging process not providing sufficient time for diffusion of Al into and rejection of Si from the κ-carbide. The lack of Si modulation observed in the 1173-10hr condition is consistent with the poorly defined isosurfaces compared to the 66hr conditions. Iron and manganese displayed strong partitioning behaviors leading to an enrichment of both in austenite for all samples. In the 1173-10hr sample, although the Mn partitioning was comparable to both of the 66hr samples, the Fe partioning was less significant. This lack of Fe partitioning suggests that the Mn partitioning is promoted by C partitioning due to the strong local interactive forces between Mn and C [21], whereas the Fe partitioning is driven by displacement or lack of Al. As a result of the large size of Si and Al, their partitioning kinetics are expected to be slower. Molybdenum showed no strong partitioning in any sample.

The proxigram data were also used to determine the composition of the total tip, austenite, and κ-carbide as shown in Table 2. The reported compositions in atomic percent (and weight percent) are the average from two tips in each processing condition. The bulk composition of the tips was generally consistent across all the conditions, except that the 1273-66hr condition had a lower Mn, Si, and Mo content than the other two conditions and higher Fe content. This decreased Mn-content may be due to unavoidable Mn-banding within the plates and is observed as an overall reduction of Mn in both the austenite and κ-carbide in this condition. As was visually observed in the proxigrams, the Al-concentration in κ-carbide is lowest in the 1173-10hr condition with the same concentration observed in the two 66hr conditions. However, silicon partitioning is less clear-cut but does appear to indicate a decrease in Si content within the κ-carbide with increasing STQ temperature. As was observed in previous APT studies, the Mo content in the κ-carbide also slightly decreased with decreasing Si content. Conversely, C concentration increased with increasing STQ temperature (decreasing Si content).

Carbon vacancy concentration of the κ-carbide was determined using a modified version of the formulation presented by Yao et al., extended to include Si and Mo [22,23]. First principles calculations have previously shown that Si preferentially occupies an Al-site in stoichiometric Fe_3_AlC and destabilizes the structure, whereas in the presence of more than 25% carbon vacancies it preferentially occupies the Fe-site and stabilizes κ-carbide, although there is still a driving force for the Si to partition to austenite [8]. Mo has also been shown to be present in κ-carbide in previous APT studies, and increases with increasing Si; however the specific site preference has not been investigated. As a result of its large atomic radius, it has been assumed to occupy the larger Al-site. Following this rationale, the formula (Fe,Mn,Al,Si,Mo)_4_(C,Va)_1_ was used to determine the carbon vacancy concentration for each condition as:-1173-10hr: (Fe_2.13_Mn_1.13_Al_0.65_Si_0.08_Mo_0.01_)(C_0.23_Va_0.77_)-1223-66hr: (Fe_2.06_Mn_1.12_Al_0.74_Si_0.06_Mo_0.01_)(C_0.23_Va_0.77_)-1273-66hr: (Fe_2.13_Mn_1.06_Al_0.75_Si_0.05_Mo_0.01_)(C_0.24_Va_0.76_)

Given the high carbon vacancy concentration Si was assumed to occupy the Fe-site, resulting in the following chemical formulas:-1173-10hr: (Fe_2.13_Mn_0.80_Si_0.08_)(Al_0.65_Mn_0.33_Mo_0.01_)(C_0.23_Va_0.77_)-1223-66hr: (Fe_2.06_Mn_0.88_Si_0.06_)(Al_0.74_Mn_0.25_Mo_0.01_)(C_0.23_Va_0.77_)-1273-66hr: (Fe_2.13_Mn_0.82_Si_0.05_)(Al_0.75_Mn_0.24_Mo_0.01_)(C_0.24_Va_0.76_)

The chemical formulas indicate that although there was a marked difference in C-content across the three conditions, there was no significant change in the total carbon vacancy concentration. The Si content decreases with increasing STQ temperature. It should also be noted that the 1273-66hr condition had a lower total Si content (−15%), which would account for the slight decrease in Si within the κ-carbide in the two 66hr conditions, thus making the Si content difference more likely a function of aging time rather than STQ temperature. The Al:Mn ratio in the Al-site for the 1173-10hr condition is −2, whereas it is −3 for the 66hr conditions. This appears consistent with the visual observation of the lack of Al-partitioning in the 1173-10hr condition but significant Mn-partitioning to the austenite in all cases. Similarly, a diminished partitioning of Fe to austenite in the 1173-10hr condition should lead to higher Fe:Mn ratio in the Fe-site; however, the 1173-10hr and 1273-10hr conditions are similarly Fe-rich (Fe:Mn −2.7), whereas 1223-66hr has a higher Mn-content (Fe:Mn −2.3). Recalling that the 1273-66hr condition tips had a lower total Mn-content and higher Fe-content, accounting for the shift in the total Fe:Mn ratio accounts would bring the κ-carbide Fe:Mn ratio in-line with the 1223-66hr condition (−2.6 Fe:Mn). Based on the chemical formulas obtained for the κ-carbide, the two points of interest are the increased Si-content and decreased Al-content (and decreased Al:Mn ratio) of the 1173-10hr condition κ-carbide and the variation in Fe:Mn ratio in the two 66hr conditions, and is shown in Table 3.

Applying the lever rule [24] to the respective phases’ concentrations measured by atom probe, the volume percent of κ-carbide for the 1173-10hr, 1223-66hr, and 1273-66hr were determined to be 42.6, 46.1, and 38.3, respectively. The ratio of the Si and Al partitioning, within the κ-carbide to the austenite as well as the amplitude difference, is calculated and shown in Table 2. The change in yield strength (ΔYS) (see Table 1) was also normalized to the volume percent of κ-carbide present, which indicated the rate of hardening from carbide precipitation is constant for the 1223-66hr and 1273-66hr conditions. It should also be noted that the amount of κ-carbide precipitated is lowest in the 1273-66hr condition, as would be expected from the age hardening rates reported in Figure 1.

## 4. Discussion

The age hardening rate of the Fe-30Mn-9Al-1Si-1C-0.5Mo lightweight steel alloy is strongly associated with the solution treatment temperature. It has been well established that age hardening of metallic alloys is typically controlled by three factors: (1) the degree of super saturation, (2) the vacancy concentration, and (3) the diffusion distance and path [25]. The lowest STQ temperature used in this study was 1173 K, which is 50 K greater than the maximum κ-carbide dissolution temperature recorded for this alloy through in situ synchrotron analysis by Ley et al. [26]. These results indicate that κ-carbide was in the solution at the STQ temperature. However, Ley et al. did note that during ex situ investigation a small amount of κ-carbide was precipitated during quenching. Vacancy concentration is not anticipated to have a significant effect on κ-carbide precipitation because the precipitation occurs from homogeneous spinodal decomposition. Diffusion path and distance remain as the only factors affecting the age hardening response of the alloy. The lowest STQ temperature (1173 K) condition has the fastest age hardening rate and finest austenite grain size at −30 μm. The slowest measured hardening response was observed after solution treating at 1323 K; this also corresponds to an austenite grain size nearly seven times larger compared to the 1173 K STQ condition (201 vs. 30 μm). Grain boundary and dislocation mediated diffusion typically require 0.5 the energy compared to bulk diffusion [27]. The 1323 K STQ materials did not exhibit any age hardening response at 773 K until aging was performed for over 20 h. The APT results further show that the volume fraction of κ-carbide precipitated during age hardening for 66hr was increased for the finer grained 1223 K STQ condition compared to the 1273-66hr condition. In this case, reducing the grain size by 33% led to a 21% increase in the precipitated κ-carbide volume fraction.

The mechanical properties of the age hardened alloy are dependent on both the starting grain size resulting from the initial STQ condition as well as the aging time. The mechanical properties of the STQ condition have previously been shown by Field et al. [19] to correlate with the grain size resulting from STQ time and temperature variations. STQ strength decreases with increasing grain size resulting from increasing STQ temperature and/or time. Since the age hardening rate also decreases with increasing grain size, the lowest temperature STQ condition with the finest grain size should have the highest starting STQ hardness/strength and age harden faster. It can then be anticipated that to achieve a target hardness level in this alloy, a range of processing conditions can be used, resulting in a range of structures, thereby affecting the overall performance. To better understand these complex processing-structure-property relationships the STQ grain size effect on mechanical properties and κ-carbide structure for age hardening the same amount of time is analyzed first, followed by the differences in mechanical response and underlying structure for two different grain sizes when age hardened to the same hardness level.

The 1223-66hr and 1273 K-66hr samples are aged for identical lengths of time, and the resulting strength increases are comparable. Both alloys appear to exhibit about an 11.5 MPa increase in strength per volume % of κ-carbide. The work by Bartlett et al. [8] showed that increasing the alloy’s Si content increased the C activity in austenite, thereby raising the C partitioning coefficient, resulting in an increased yield strength that was well described by Kato et al. [28]. Prediction of the strength increase due to κ-carbide precipitation can be predicted using the methodology presented by Kato et al., and is shown by Equations (2) and (3).
(2)$$\Delta \tau_{Y}=\frac{\epsilon }{\sqrt{6}}\left(C_{11}-C_{12}\right)$$
(3)$$\epsilon =\frac{A\eta \left(C_{11}+2C_{12}\right)}{3C_{11}}$$
where Δ*τ_Y_* is the change in the shear stress, *ϵ* is the strain amplitude, *A* is the concentration amplitude of carbon measured to be 4.8 and 4.7 at. % as measured between the min and max *C* content associated with the κ and γ phases for the 1223-66hr and 1273-66hr, respectively. The matrix elastic constants *C*_11_ and *C*_12_ are set as 14.4 × 10^10^ and 8.7 × 10^10^ N/m^2^, respectively, according to the work by Sato et al. [29]. The *η* term is the change in the lattice parameter with increasing carbon content and was determined by Sato et al. [29] to be 0.18, multiplying by the Taylor factor of 3.06 for fcc metals, gives the change in yield strength from spinodal hardening. This analysis yields a strength increase of 453 and 447 MPa for the 1223-66hr and 1273-66hr samples, respectively. This is a good estimation of the strength increase for the 1273-66hr condition (435 MPa), but underestimates the increase observed in the 1223 K-66hr condition (560 MPa). This may be because the Kato analysis assumes an equal volume fraction of κ-carbide is precipitated. Taking into account the increased volume of κ-carbide precipitated (46 vs. 38 vol. %) produces an additional strength increase of 93 MPa and would explain the discrepancy between the measured strength increase in the 1223-66hr to the calculated strength increase.

The 1173-10hr and 1273-66hr conditions have similar hardness (323 ± 9 vs. 353 ± 5 HBW), but different underlying structures, resulting in different overall mechanical performance. The yield strengths after age hardening are similar (850 ± 10 vs. 875 ± 8 MPa); however, the strength increases due to κ-carbide precipitation are drastically different. The 1173-10hr condition only exhibits an increase in strength of −300 MPa and contains 42.6 vol. % κ-carbide. The 1273-66hr condition exhibited a significantly greater increase in strength of 440 MPa with a lower κ-carbide content of 38.3 vol. %. It was noted earlier that a greater amount of κ-carbide is formed after a shorter time due to the smaller grain size; yet, since the strengths are the same, it is theorized that a greater portion of the strength is due to traditional Hall-Petch strengthening reported previously by the authors [19] and shown in Equation (4).
(4)$$\sigma_{H-P}\left(\mathrm{MPa}\right)=\frac{1244\left(\mathrm{MPa}\sqrt{\mu m}\right)}{\sqrt{L_{3}\left(\mu m\right)}}+332$$

A second consideration is the difference in chemical partitioning observed in Figure 7 for these conditions. The 1273-66hr sample exhibits the typical modulated structure of carbon and aluminum partitioning to the κ-carbide with partitioning coefficients of 2.21 and 1.14, respectively, and silicon being rejected into the austenite with a partitioning coefficient of 0.65. The 1173-10hr sample has a comparable carbon partitioning coefficient of 2.04, though the silicon and aluminum profiles are relatively flat, with partitioning coefficients of 0.98 and 0.94, respectively, indicating no significant partitioning. The shorter aging time appears to have been insufficient for Al diffusion to the κ-carbide and Si diffusion to the austenite to occur, leaving the profile relatively unchanged from the super saturated condition. This implies that the effects of aluminum and silicon amplitude cannot be ignored and provide a significant strengthening effect to the alloy. The assumption of Si partitioning to austenite during precipitation would increase the matrix strength through solid solution strengthening; however, in the case of the 1173-10hr condition there is no increase in Si and no decrease in Al in the austenite. The effect of Si and Al partitioning on the resulting strengthening during κ-carbide precipitation was considered as the change in yield strength from the STQ and aged conditions, and the total volume percent κ-carbide measured by APT as shown in Figure 9. A strong correlation is obtained for all three APT analyzed samples, indicating that the partitioning of silicon to the austenite and aluminum to κ-carbide provides a significant contribution to strength that has not been previously addressed. Bartlett et al. [8] and Wang et al. [20] showed the effect of total silicon on the strength and ductility. However, the alloys were heat treated at the same times and temperatures, and thus no change in the silicon modulation was noted between alloys or conditions. Consequently, this work illustrates the unique effect of Si partitioning on the strength of lightweight steels.

The work hardening behavior of age hardenable lightweight steel has been noted by Choi et al. [11] to be increased in the under-aged condition compared to material that is over-aged. This hypothesis would appear to be in direct contrast to the results presented in this work, where the 1173-10hr condition has both a higher work hardening rate and volume fraction of κ-carbide relative to the 1273-66hr condition. This implies that the 1273-66hr sample is under-aged relative to the 1173-10hr sample, yet it exhibits a lower work hardening rate. However, this neglects the significant differences in chemical partitioning between these two conditions. In the case of the shorter aging time, the lesser degree of Al and Si partitioning produces a steel with both higher strengths and improved work hardening capability. This has also been noted in alloys where the Si concentration is varied at fixed aging times and temperatures [14,20]. The alloys with lower silicon content exhibited greater work hardening, yet both studies also showed that the lower Si content produced less Si partitioning, while simultaneously precipitating less κ-carbide. Interestingly, this work shows that, by reducing the Si modulation, an improvement in work hardening is obtained even with an increased volume fraction of κ-carbide present.

The improvement in work hardening is integral to providing the greatest energy absorption. The greater work hardening condition provides more area under the curve with an improved yield strength 850 MPa. This is consistent with the results presented by Ley et al. [26]. They showed through synchrotron x-ray diffraction that the toughness, as measured by Charpy v-notch impact testing, was doubled without a loss of strength when the heat treatment was designed, such that sideband formation was reduced. The observed sidebands were shown to occur when Al and C were rejected from the austenite into the κ-carbide, producing a lattice contraction of the austenite. This lattice contraction is further exacerbated when Si is forced into the austenite from the κ-carbide. Ley et al. noted that the best performing condition had the lowest sideband diffractograms. It is postulated from this work that the reduction in sideband was due to a reduction in Si partitioning, while also providing the increase in toughness to the steel.

## 5. Conclusions

From this work, it can be concluded that the partitioning of Si and Al play a significant role in the mechanical response of the κ-carbide strengthened lightweight steels. It was found that by reducing the solution treatment temperature and retaining a finer grain size, the aging hardening response of the alloy was significantly accelerated, producing hardening after 1hr at 773 K. From the reduced aging time, the diffusion of Si and Al were reduced, and APT results confirmed a smaller degree of Si and Al partitioning between the κ-carbide and austenite. The reduced Si and Al modulation gave rise to a significant increase in the work hardening without a loss in the yield strength or total ductility; consequently producing a very high tensile strength (>1460 MPa) without sacrificing ductility. Thus, the results confirm through proper heat treatment control, these lightweight steels can produce improved energy absorption without sacrificing strength.

## Figures and Tables

**Figure 1 materials-15-01670-f001:**
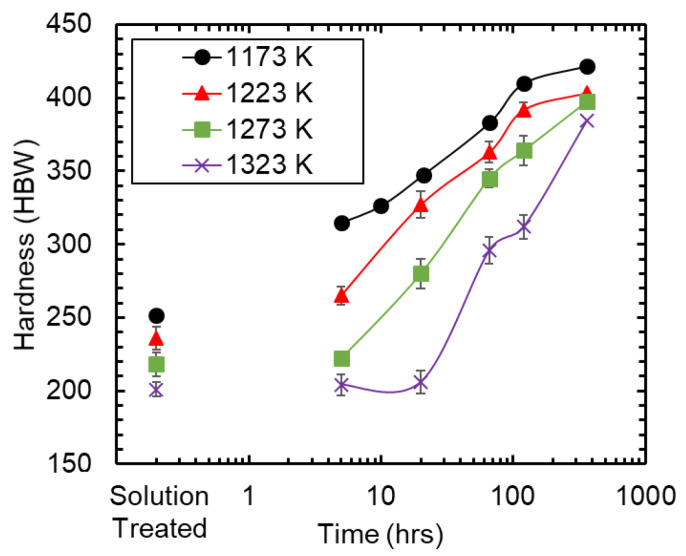
Age hardening response of the Fe-Mn-Al steel after various solution treatment conditions aged at 773 K.

**Figure 2 materials-15-01670-f002:**
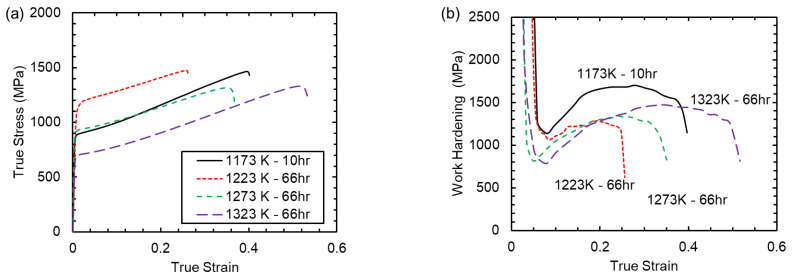
(**a**) Stress-strain graph of the samples aged at 773 K. (**b**) Instantaneous work hardening rate of the alloy after varying heat treatments.

**Figure 3 materials-15-01670-f003:**
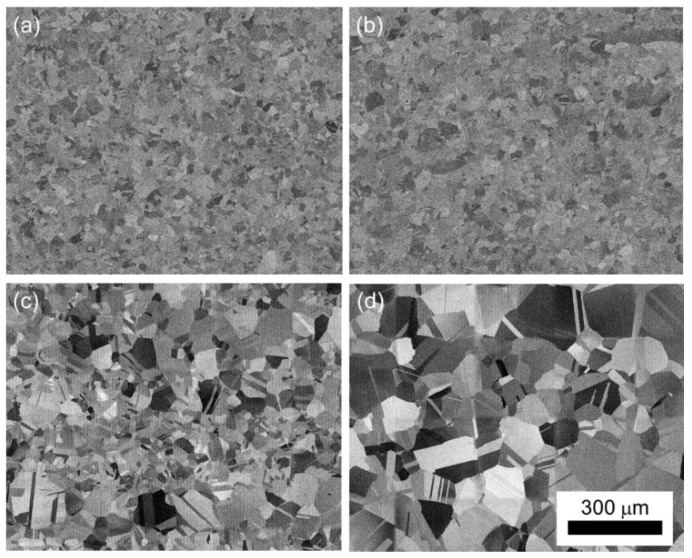
Scanning electron micrograph of the alloy after solution treatment at (**a**) 1173 K, (**b**) 1223 K, (**c**) 1273 K, and (**d**) 1323 K for two hours followed by water quenching. All images taken at the same magnification.

**Figure 4 materials-15-01670-f004:**
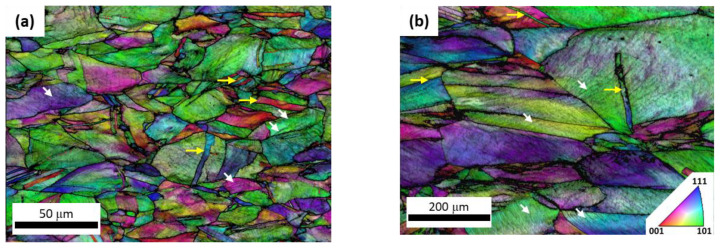
Orientation image mapping (OIM) of the tensile tested alloys in the (**a**) 1173-10hr, (**b**) 1323-66hr conditions. The axis of deformation is parallel to the horizontal of the image. Slip bands are identified with white arrows and annealing twins are shown with yellow arrows.

**Figure 5 materials-15-01670-f005:**
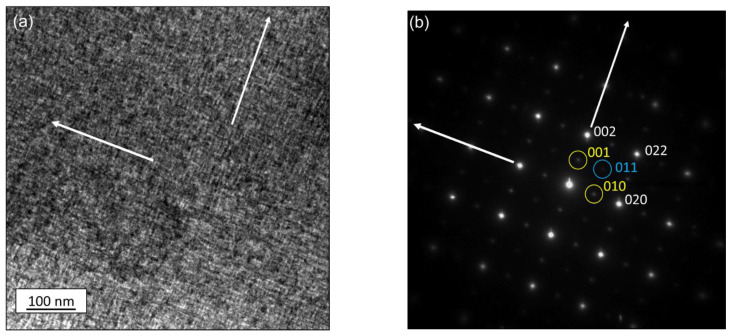
(**a**) TEM image of the 1223-66hr sample Grain aligned at (**b**) <001> zone axis shows alignment of spinodal decomposition predominantly along 002; superlattice reflections are evident at <001> (primary-outlined in yellow) and 011 (secondary-outlined in blue).

**Figure 6 materials-15-01670-f006:**
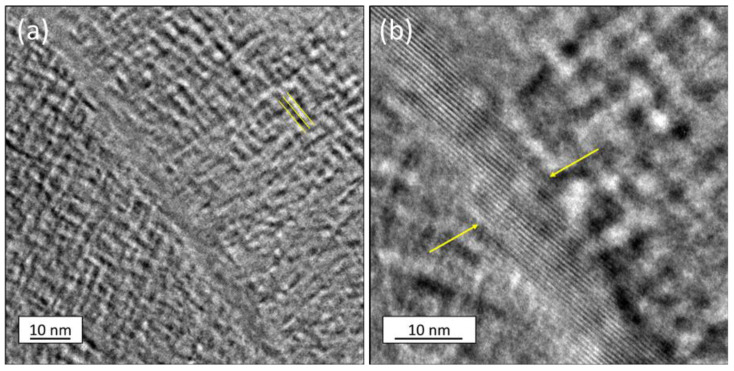
(**a**) TEM image of the 1173-10hr with yellow lines showing spinodal decomposition wavelength along <001> of −2–3 nm (**b**) Higher resolution of grain boundary with 10 nm FeMnAl phase at low-angle grain boundary, facets of grain boundary terminate with spinodal orientation on each on each side.

**Figure 7 materials-15-01670-f007:**
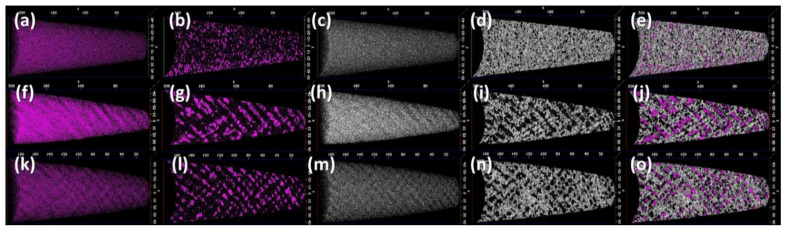
Atom map and isoconcentration surfaces of the full atom probe tip for samples (**a**–**e**) 1173-10hr (**f**–**j**) 1223-66hr, and (**k**–**o**) 1273-66hr. Carbon atom maps (**a**,**f**,**k**) and isoconcentration surfaces are shown for each sample condition as (**b**) 5.60 at. %, (**g**) 5.55 at. %, and (**l**) 5.74 at. %. Silicon atom maps (**c**,**h**,**m**) and isoconcentration surfaces are shown for each sample condition as (**d**) 1.93 at %, (**i**) 2.27 at. %, and (**n**) 1.88 at. %. Composite images of the Si and C isoconcentration surfaces are shown in (**e**,**j**,**o**) for each sample condition.

**Figure 8 materials-15-01670-f008:**
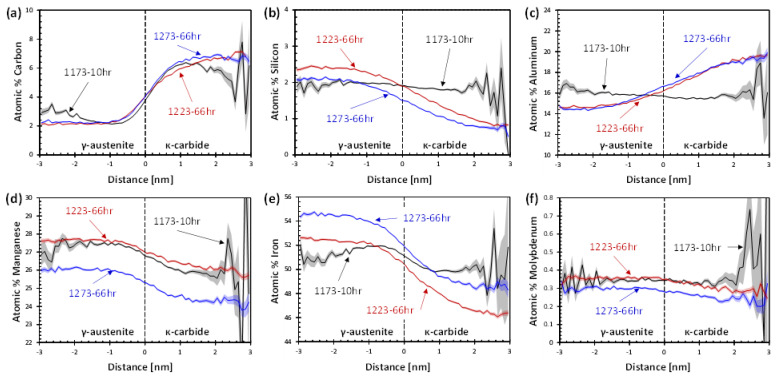
Proximity histograms displaying representative concentration profile (average line and error shaded region) across the interface of austenite and κ-carbide for (**a**) carbon and (**b**) silicon, (**c**) aluminum, (**d**) manganese, (**e**) iron, and (**f**) molybdenum.

**Figure 9 materials-15-01670-f009:**
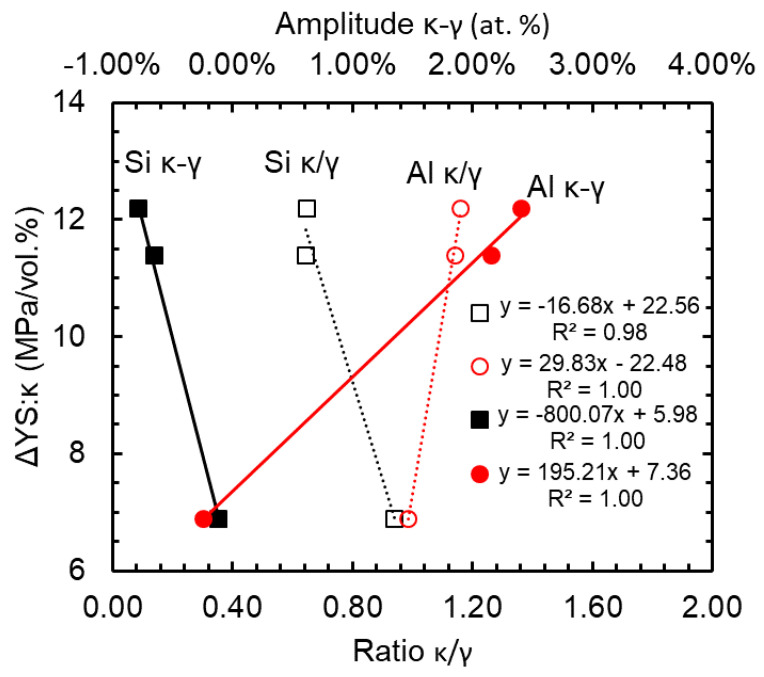
Effect of silicon amplitude on the strength increment increase with κ-carbide precipitation.

**Table 1 materials-15-01670-t001:** Microstructural and mechanical behavior of the alloy investigated.

Solution Temperature (K)	Grain Size (μm) *	Condition	Hardness (HBW)	Yield Strength (MPa)	True Tensile Strength (MPa)	True Strain	ΔYS (MPa)
1173	30 ± 2	STQ *	245 ± 7	555 ± 7	1499 ± 9	0.49 ± 0.02	-
		773 K-10hr	332 ± 9	850 ± 10	1464 ± 13	0.40 ± 0.01	290
1223	57 ± 8	STQ *	236 ± 9	505 ± 7	1504 ± 7	0.55 ± 0.01	-
		773 K-66hr	385 ± 10	1060 ± 3	1472 ± 7	0.26 ± 0.03	555
1273	156 ± 6	STQ *	224 ± 3	440 ± 15	1481 ± 2	0.59 ± 0.01	-
		773 K-66hr	353 ± 5	875 ± 8	1316 ± 10	0.36 ± 0.03	435
1323	201 ± 8	STQ *	201 ± 2	410 ± 11	1429 ± 6	0.61 ± 0.01	-
		773 K-66hr	246 ± 10	650 ± 9	1330 ± 11	0.52 ± 0.02	240

* Data taken from ref. [19].

**Table 2 materials-15-01670-t002:** Chemical composition in atomic % (weight %) of the total tip, austenite, and κ-carbide determined by atom probe tomography for each of the aged conditions.

	Total			γ-Austenite			κ-Carbide		
	1173-10hr	1223-66hr	1273-66hr	1173-10hr	1223-66hr	1273-66hr	1173-10hr	1223-66hr	1273-66hr
Fe	50.96 ± 0.18	50.37 ± 0.19	52.16 ± 0.29	51.59 ± 0.15	52.02 ± 0.26	53.50 ± 0.26	50.11 ± 0.28	48.47 ± 0.26	50.04 ± 0.23
	(58.25 ± 0.21)	(57.81 ± 0.22)	(59.76 ± 0.33)	(58.41 ± 0.17)	(58.69 ± 0.29)	(60.43 ± 0.30)			
Mn	27.10 ± 0.33	27.03 ± 0.05	25.52 ± 0.11	27.49 ± 0.31	27.51 ± 0.03	25.96 ± 0.11	26.57 ± 0.31	26.47 ± 0.02	24.84 ± 0.13
	(30.48 ± 0.37)	(30.52 ± 0.05)	(28.77 ± 0.12)	(30.62 ± 0.34)	(30.54 ± 0.03)	(28.84 ± 0.13)			
Al	15.54 ± 0.22	16.17 ± 0.23	16.34 ± 0.18	15.64 ± 0.24	15.05 ± 0.27	15.51 ± 0.18	15.40 ± 0.20	17.45 ± 0.27	17.67 ± 0.13
	(8.58 ± 0.12)	(8.97 ± 0.13)	(9.05 ± 0.10)	(8.56 ± 0.13)	(8.21 ± 0.15)	(8.46 ± 0.10)			
C	3.85 ± 0.12	3.89 ± 0.09	3.81 ± 0.03	2.67 ± 0.13	2.49 ± 0.04	2.59 ± 0.01	5.44 ± 0.22	5.51 ± 0.03	5.72 ± 0.02
	(0.95 ± 0.03)	(0.96 ± 0.02)	(0.94 ± 0.01)	(0.65 ± 0.03)	(0.60 ± 0.01)	(0.63 ± 0.00)			
Si	1.89 ± 0.03	1.88 ± 0.01	1.59 ± 0.04	1.94 ± 0.02	2.25 ± 0.02	1.85 ± 0.04	1.82 ± 0.04	1.46 ± 0.03	1.19 ± 0.02
	(1.09 ± 0.02)	(1.09 ± 0.01)	(0.92 ± 0.02)	(1.11 ± 0.01)	(1.28 ± 0.01)	(1.05 ± 0.02)			
Mo	0.33 ± 0.01	0.33 ± 0.01	0.29 ± 0.00	0.34 ± 0.01	0.35 ± 0.01	0.30 ± 0.00	0.33 ± 0.01	0.31 ± 0.01	0.26 ± 0.00
	(0.66 ± 0.02)	(0.66 ± 0.02)	(0.56 ± 0.00)	(0.66 ± 0.02)	(0.68 ± 0.02)	(0.58 ± 0.00)			

**Table 3 materials-15-01670-t003:** Data of interest for further analysis including: Si and Al partitioning ratio and amplitude from APT proxigrams, vol. % κ from APT, included is the ratio of the strength increase as related to the κ-carbide volume content (ΔYS:κ).

STQ Temperature [K]	Age Time [hr]	Si Ratio (κ/γ)	Si Amplitude [at. %] (κ-γ)	Al Ratio (κ/γ)	Al Amplitude [at. %] (κ-γ)	Vol. % κ	ΔYS:κ [MPa/vol. %]
1173	10	0.94	−0.12	0.98	−0.24	42.6	6.9
1223	66	0.65	−0.79	1.16	2.40	46.1	12.2
1273	66	0.64	−0.66	1.14	2.16	38.3	11.4

## Data Availability

Not applicable.

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
