# Peer review of "Alloy Partitioning Effect on Strength and Toughness of κ-Carbide Strengthened Steels"

_materials, 2022, doi:10.3390/ma15051670_

Round 1
Reviewer 1 Report
This paper pays attention to the partitioning effect of Si and Al on the mechanical response of the κ-carbide strengthened lightweight steels. It is an interesting topic, and some valuable conclusions have been achieved. A minor revise is needed.
1)The experimental schedule and the schedule purpose should be described in the whole at Section2.
2)Author mentioned that 66 hours follow STQ at 1223, 1273, and 1323 K. Why only one time of 66 hours was adopted for different temperatures?
3)The curves of instantaneous work hardening rate in Fig.2b are very strange as some curves are serrated. Reviewer doubt these serrated curve were not calculated from the stress-strain curves in Fig.2a.
4)The yield strength from the STQ in Fig.9 should be defined in details.
Author Response
Reviewer #1
This paper pays attention to the partitioning effect of Si and Al on the mechanical response of the κ-carbide strengthened lightweight steels. It is an interesting topic, and some valuable conclusions have been achieved. A minor revise is needed.
Thank you for the favorable review and positive comments, adjustment have been made to improve upon the clarity and presentation of this work.
1)The experimental schedule and the schedule purpose should be described in the whole at Section2.
The rational for the testing and investigation was incorporated to the text. "The 1223, 1273 and 1323 K solution treatment followed by a 66hr age hardening treatment were selected to investigated the effect of a fixed aging time on microstructure and material behavior. The 1173 K STQ treatment was followed by 10hr age hardening to compare the effect of a variable age hardening time to a fixed strength."
2)Author mentioned that 66 hours follow STQ at 1223, 1273, and 1323 K. Why only one time of 66 hours was adopted for different temperatures?
The authors were attempting to investigate the effects at either a fixed strength, as well as at a fixed time. This required the combination of the 1173K-STQ aging to be performed at a shorter time. The 1323K however, was slightly softer than anticipated
3)The curves of instantaneous work hardening rate in Fig.2b are very strange as some curves are serrated. Reviewer doubt these serrated curve were not calculated from the stress-strain curves in Fig.2a.
The serrations observed are from minute changes in the load associated with noise in tracking the data. The data was shown without smoothing to show the full behavior. For simplicity and to reduce confusion some smoothing will be performed to reduce the noise in the work-hardening curves.
Figure 2(b) has been updated and the text has been amended to note the smoothing “logarithmic smoothing”
4)The yield strength from the STQ in Fig.9 should be defined in details.
A note is provided for a reference back to Table I for the exact values
Reviewer 2 Report
The manuscript titled “Alloy Partitioning Effect on Strength and Toughness of Light-2 weight Steels” is interesting but lacks coherence in interpretation of the results. The experiments are well planned but a few of them lack a strong analysis of the data. This leaves the reader confused about a lot of aspects of the work. I would therefore recommend this work for publication in Materials after intensively modifying the manuscript. Detailed reasons are given below:
- One issue for the manuscript is the unclearness of the motivation in the introduction and the significance of the findings.
- The authors should improve the discussion of presented results and include a comparison between their results and other computational and/or experimental works where possible
- All the SEM figure needs to have scale bar
- What’s the grain size of the all the samples?
- What is the reason of the stress hardening rate is improving after the 1223K-66h?
Reviewer 3 Report
The manuscript is very interesting and the involved methods provide a good insight into a role of k-carbide in solution-treated FeMnAlSiMo steel. Especially, the intresting result is determination of carbon vaccancy concentration and position of individual elements in the carbides.
However, I would mark two situations that could be improved to better understand the effect of heat-treatment of lightweight steels on their microstructure and properties that are communicated in the manuscript.
Firstly, the role of presenting EBSD maps for two extreme situations is not fully described. I see in the microstructures in Fig. 3 that the grain size is totally different, but why have you decided to present those two EBSD maps in the manuscript. Besides, the scale in the images is different - again, why have you chosen so? what is important in those images that you aim to highlight. Moreover, the authors have stated "strain is accomodated by slip not twinning or phase transformation" - it is not clear by looking at the images in Figure 4. Perhaps you could mark the slip marks or provide a missorientation map (of local or intragranular missorientations).
Secondly, it is very important to follow the position of elements across austenite/k-carbide interface. Therefore, it seems to be a good custom to insert the name of element into the proximity histograms to follow the text easier. The analysis of histograms is important to understand the mechanisms involving k-carbide precipitation in austenite, thus it is important to clearly read graphs in Fig. 8.
I would also recomend the authors to include the term k-carbide in the title so the work is better searchable. Basically the whole paper deals with the kappa-carbides. However, this is just a suggestion, of a minor importance.
Some additional remarks:
line 84: there is an acronym SFE, it should be explained as in other situations throught the manuscript. Later the authors use the acronym instead the full name in line 176
line 334: the sentence "Ley et al however did note that during ex situ¬¬ ..." is confusing; is there something missing or there should be some symbol?
line 341: a sentence "The slowest age hardening rate for the 1323 K STQ condition and has a nearly 7 times larger grain size" - seems to be incomplete
line 370: "Kato et al.’s" - I am not sure if this is correct?
line 448: "syncrotron" is misspelled
References: the format of citation should be unified and font should be the same
Author Response
Reviewer #2
The manuscript is very interesting and the involved methods provide a good insight into a role of k-carbide in solution-treated FeMnAlSiMo steel. Especially, the intresting result is determination of carbon vaccancy concentration and position of individual elements in the carbides.
Thank you for your timely response on the paper. The authors are pleased to know that this work was found interesting and valuable. Thank you,
However, I would mark two situations that could be improved to better understand the effect of heat-treatment of lightweight steels on their microstructure and properties that are communicated in the manuscript.
Firstly, the role of presenting EBSD maps for two extreme situations is not fully described. I see in the microstructures in Fig. 3 that the grain size is totally different, but why have you decided to present those two EBSD maps in the manuscript. Besides, the scale in the images is different - again, why have you chosen so? what is important in those images that you aim to highlight. Moreover, the authors have stated "strain is accomodated by slip not twinning or phase transformation" - it is not clear by looking at the images in Figure 4. Perhaps you could mark the slip marks or provide a missorientation map (of local or intragranular missorientations).
EBSD figures were incorporated to identify that the material is in fact a SLIP dominated deformation steel, consistent to what has been previously reported in the literature for SFE>45 mJ/m2. The difference in magnification was necessitated to show enough grains for both images.
To identify key microstructural features Figure 4 has been updated. Text has also been added to describe the features in the EBSD image. “Slip bands are indicated with white arrows and the annealing twins, not mechanical twins, are identified by yellow arrows. It should be noted that the annealing twin boundaries are no longer parallel due to the deformation applied during tensile testing, and boundaries appear bent or bowed.”
Secondly, it is very important to follow the position of elements across austenite/k-carbide interface. Therefore, it seems to be a good custom to insert the name of element into the proximity histograms to follow the text easier. The analysis of histograms is important to understand the mechanisms involving k-carbide precipitation in austenite, thus it is important to clearly read graphs in Fig. 8.
This is true and the authors appreciate the comment to improve upon the clarity of the presentation of the paper. Figure 8 contains axis labels for every graph with unique titles indicating the element of interest as well as the matrix vs κ-carbide.
I would also recommend the authors to include the term k-carbide in the title so the work is better searchable. Basically the whole paper deals with the kappa-carbides. However, this is just a suggestion, of a minor importance.
The title has been modified to: “Alloy Partitioning Effect on Strength and Toughness of κ-caribide Strengthened Steels”
Some additional remarks:
line 84: there is an acronym SFE, it should be explained as in other situations throught the manuscript. Later the authors use the acronym instead the full name in line 176
This has been corrected and addressed
line 334: the sentence "Ley et al however did note that during ex situ¬¬ ..." is confusing; is there something missing or there should be some symbol?
Thank you, this is a typographical error due to copying into the Materials format and was not caught. The text has been corrected to: “Ley et al however did note that during ex situ investigation a small amount of κ-carbide was precipitated during quenching”
line 341: a sentence "The slowest age hardening rate for the 1323 K STQ condition and has a nearly 7 times larger grain size" - seems to be incomplete
This sentence has been adjusted to the following to improve upon the clarity: “The slowest measured hardening response was observed after solution treating at 1323 K this also corresponds to an austenite grain size nearly 7 times larger compared to the 1173 K STQ condition (201 vs 30 μm).”
line 370: "Kato et al.’s" - I am not sure if this is correct?
This has been adjusted to “Prediction of the strength increase due to κ-carbide precipitation can be predicted using the methodology present by Kato et al. and is shown by…”
line 448: "syncrotron" is misspelled
Thank you the spelling has been checked and proofed again.
References: the format of citation should be unified and font should be the same
References and font have been made consistent
Reviewer 4 Report
The article entitled “Alloy Partitioning Effect on Strength and Toughness of Light-2 weight Steels” authored by Field et al. reports the effect of solution treatment on Al, Si alloy partitioning and mechanical properties of light weight steel. The article has 9 figures, 3 tables and 31 references. The qualities of figures are good. The authors established their claims with the aid of sophisticated equipment's such as SEM, TEM, EBSD, OIM, Atom mapping. Though the authors are appreciated for undertaking an experimental study, the following queries needs to be addressed before considering for publication.
- The present work seems to be a continuation of a previous study [Ref.19]. The effect of solution hardening on grain size and strength are already reported on the article.
- Authors must expand the Acronyms used in the manuscript at the first instant of usage.
- Title does not convey the work carried out in the present study. It is recommended to modify the title.
- More emphasis on presenting the novelty of the work is recommended.
- Authors need to state how the age hardening temperature and duration were fixed.
- Authors selected one condition for each STQ condition, based on hardness. It is recommended to compare results in a common platform.
- Authors are directed to mark the salient features in the microstructure itself for better understanding.
- The significance of k-carbides are not discussed only in the lateral part of the manuscript.
- Authors reported the tensile strength for the same conditions in Ref.19, and repeated the same in the present study.
Author Response
The article entitled “Alloy Partitioning Effect on Strength and Toughness of Light-2 weight Steels” authored by Field et al. reports the effect of solution treatment on Al, Si alloy partitioning and mechanical properties of light weight steel. The article has 9 figures, 3 tables and 31 references. The qualities of figures are good. The authors established their claims with the aid of sophisticated equipment's such as SEM, TEM, EBSD, OIM, Atom mapping. Though the authors are appreciated for undertaking an experimental study, the following queries needs to be addressed before considering for publication.
Thank you for your appreciation of the work performed, the authors have worked to address all of the concerns raised by the reviewer.
- The present work seems to be a continuation of a previous study [Ref.19]. The effect of solution hardening on grain size and strength are already reported on the article.
This work was done in conjunction with the previous report on grain size. However the purpose of this work investigated the effect of κ-carbide and how element partitioning is very closely related to the incumbent austenite grain size. This is novel with respect to what is typically understood and had a larger effect on the developed properties. This was separated from them previous work because the previous paper deal purely with the kinetic of grain growth on this steel and effect on SFE.
- Authors must expand the Acronyms used in the manuscript at the first instant of usage.
This has been done to the best of the authors ability
- Title does not convey the work carried out in the present study. It is recommended to modify the title.
The title has been modified to: “Alloy Partitioning Effect on Strength and Toughness of κ-caribide Strengthened Steels”
- More emphasis on presenting the novelty of the work is recommended.
The authors appreciate that it was noted the work was novel. The concluding remark in the introduction states: “All previous works have looked at the effect of either a fixed time of aging with varying age hardening temperature or a fixed aging condition (time and temperature) for a variation of compositions. This work is novel in that it will investigate the interacting effects of solution treatment of a single alloy on the mechanical behavior are related to the compositional response of the alloy at the nano-carbide level.”
- Authors need to state how the age hardening temperature and duration were fixed.
Age hardening times are further explained in the methods section.
- Authors selected one condition for each STQ condition, based on hardness. It is recommended to compare results in a common platform.
The methodology is further explained, and it should be noted that due to this approach the results were first discovered and noted in this work
- Authors are directed to mark the salient features in the microstructure itself for better understanding.
This is done through-out the paper on Figures 4 and 8 to better identify key components of the microstructure.
Text has also been added to describe the features in the EBSD image. “Slip bands are indicated with white arrows and the annealing twins, not mechanical twins, are identified by yellow arrows. It should be noted that the annealing twin boundaries are no longer parallel due to the deformation applied during tensile testing, and boundaries appear bent or bowed.”
- The significance of k-carbides are not discussed only in the lateral part of the manuscript.
The work is noted to provide insight into the role of k-carbide in solution-treated FeMnAlSiMo steel. Especially, the interesting result is determination of carbon vacancy concentration and position of individual elements in the carbides.
- Authors reported the tensile strength for the same conditions in Ref.19, and repeated the same in the present study.
The data was taken from Ref 19 to elucidate the effect of the change in yield strength between the as-solution treated condition and the change in strength after age hardening. The authors did not wish to double report data with respect to presenting the information that was previously report as if it was a new and unreported result. This is noted in a superscript and referenced accordingly
Reviewer 5 Report
In this article alloy partitioning effect on strength and toughness of light weight steels is investigated. Authors applied various characterisation techniques to analysis the phenomenon. The tackled research problem is interesting, however, it demands some additional comments.
- Materials and methods: There is no information what FIB microscope was used and what parameters of sputtering were applied (company, accelerating voltage, currents for final sample preparation).
- In Table 1 there is no standard deviation of a given grain size. Additionally, there is true strain but in the experimental total elongation is mentioned. It should be normalised.
- Have authors tried to do FFT of Fig. 6a) and b)?
- What is the twinning frequency for various conditions? What is the impact of twin boundaries on the investigated phenomenon?
- In the sentence “As is expected due to the alloy’s very high stacking fault energy, strain is accommodated by slip not twinning or phase transformation.” it is written that this alloy is characterised by very high stacking fault energy. However, it is also written that the SFE of this material is 89 mJ/m2, which his not high SFE. Additionally in line 84, the abbreviation SFE has been introduced without the explanation of its meaning, whereas in line 176 the phrase “stacking fault energy” is used without abbreviation. It should be corrected.

Author Response
In this article alloy partitioning effect on strength and toughness of light weight steels is investigated. Authors applied various characterisation techniques to analysis the phenomenon. The tackled research problem is interesting, however, it demands some additional comments.
- Materials and methods: There is no information what FIB microscope was used and what parameters of sputtering were applied (company, accelerating voltage, currents for final sample preparation).
The section of FIB preparation has now been incorporated into the text in the Materials and Methods: “FIB samples were taken from grain boundaries using a ThermoFisher Scientific Helios G4 UX dual-beam focused ion beam (FIB) / scanning electron microscope (SEM). The FIB was operated at 30 keV with maximum and minimum beam currents of 0.75 nA and 90 pA, respectfully. To minimize Ga implantation into the tip, the final step was conducted at 5 keV with a beam current of 21 pA”
- In Table 1 there is no standard deviation of a given grain size. Additionally, there is true strain but in the experimental total elongation is mentioned. It should be normalized.
The grain sizes were taken from the work of Ref 19 and the standard deviations are given in that paper. The authors did not incorporate the standard deviation for simplicity however these values are reported previously, but have been added to this text for simplicity.
The total elongation/true strain statements have been unified to all being true strain.
- Have authors tried to do FFT of Fig. 6a) and b)?
The authors did do Fast Fourier Transforms however it did not provide an increase in the value of this work so it was omitted
- What is the twinning frequency for various conditions? What is the impact of twin boundaries on the investigated phenomenon?
The material due to its SFE does not exhibit mechanical twinning. The impact of annealing twin boundaries was not investigated in this work and twin boundaries were not considered.
- In the sentence “As is expected due to the alloy’s very high stacking fault energy, strain is accommodated by slip not twinning or phase transformation.” it is written that this alloy is characterized by very high stacking fault energy. However, it is also written that the SFE of this material is 89 mJ/m2, which his not high SFE. Additionally in line 84, the abbreviation SFE has been introduced without the explanation of its meaning, whereas in line 176 the phrase “stacking fault energy” is used without abbreviation. It should be corrected.
It is well documented that when the SFE > 45 mJ/m2 an austenitic steel will accommodate deformation through slip. This is not the purpose of the work however it is noted for proper documentation that this alloy is a slip type steel.
The word “very” has been omitted however the SFE is high and is considered a slip steel as denoted above.
The term SFE/Stacking fault energy has been corrected such that upon first introduction it states “..stacking fault energy (SFE)…” and further described by the acronym SFE.
Reviewer 6 Report
Comments are in the attached file

Author Response
Thank you for the comment. The true strain was measured in-situ and as such is an accurate representation of the true stress-true strain. The cause for true stress-strain curves lacking in post necking response is that it is a calculation and cannot take into account the reduction in area. Because this investigation utilized DIC a measurement of the gauge was performed in-situ and allowed for full field strain and stress measurement and localization.
Round 2
Reviewer 4 Report
The authors responded to the reviewer comments positively. Hence, the article is recommended for publication.
Author Response
Thanks for your valuable comments.